# Performance Improvement of an STS304-Based Dispensing Needle via Electrochemical Etching

**DOI:** 10.3390/mi14122183

**Published:** 2023-11-30

**Authors:** Yong-Taek Kwon, Sanghyun Jeon, Jun Lee, Juheon Kim, Sangmin Lee, Hyungmo Kim

**Affiliations:** 1School of Mechanical Engineering, Gyeongsang National University, Jinju 52828, Republic of Korea; kyt7905@gnu.ac.kr (Y.-T.K.); 2018011906@gnu.ac.kr (S.J.); sk30101234@gnu.ac.kr (J.L.); 2School of Mechanical and Aerospace Engineering, Gyeongsang National University, Jinju 52828, Republic of Korea; juheon9827@naver.com; 3Division of Mechanical, Automotive, Robot Component Engineering, Dong-Eui University, Busan 47340, Republic of Korea

**Keywords:** STS304, microstructure, porous structure, droplet, dispensing needle

## Abstract

In this study, we explored the formation of micro-/nanosized porous structures on the surface of a needle composed of STS304 and examined the effect of conventional needles and needles capable of liquid ejection. Aqua regia, composed of HCl and HNO_3_, was electrochemically etched to form appropriately sized micro-/nanoporous structures. We observed that when dispensing liquids with low surface tension, they do not immediately fall downward but instead spread over the exterior surface of the needle before falling. We found that the extent of spreading on the surface is influenced by an etched porous structure. Furthermore, to analyze the effect of surface tension differences, we dispensed liquids with varying surface tensions using etched needles. Through the analysis, it was confirmed that, despite the low surface tension, the ejected droplet volume and speed could be stably maintained on the etched needle. This indicates that the spreading phenomenon of the liquid on the needle surface just before ejection can be controlled by the micro/nanoporous structure. We anticipate that these characteristics of etched needles could be utilized in industries where precision dispensing of low-surface-tension liquids is essential.

## 1. Introduction

Enhancing the efficiency of material surfaces has been increasingly researched based on various processing methods. One method for modifying material surfaces involves the creation of porous structures, which can be formed using multiple techniques, including electrochemical etching [1,2]. This process facilitates the creation of structural features on the material surface without significantly damaging the original state of the material [3]. Therefore, several etching methods have been explored to form porous structures [4,5]. 

The porous structures formed through etching offer various advantages by efficiently allowing substances, whether large or small, to pass through [6,7,8,9]. This property plays a crucial role in diverse applications, such as controlling gases or liquids passing through structures of specific sizes by creating fine, porous structures. This capability provides practical use in industry for processes like filtering and selective separation [10,11]. Moreover, porous structures formed through etching contribute to an enhanced surface area, thereby increasing energy exchange efficiency. This is particularly significant for energy storage devices and electronic devices. The expanded surface area facilitates chemical reactions and interactions between substances, promoting improved efficiency in energy conversion. These characteristics present opportunities for contributions to emerging energy technologies and environmentally friendly practices [12,13].

Both large and small structures are known to modify the wettability on the surface of a material [14,15,16,17]. Wettability is one of several characteristics that describe the interaction between a liquid and a solid surface [18,19]. It is distinguished by the contact angle in static conditions and the advancing or receding angles of the liquid as it moves along the surface in dynamic conditions. These angles serve as scales to differentiate between hydrophilicity and hydrophobicity, playing a crucial role in understanding the surface properties of materials. The static contact angle reflects how well a liquid spreads or beads up on a solid surface in a stable state. Hydrophilic surfaces exhibit good wetting, resulting in a low contact angle, while hydrophobic surfaces cause the liquid to maintain a distinct shape with a higher contact angle. The dynamic advancing and receding angles describe the behavior of the liquid as it moves along the surface. These characteristics influence factors such as the ability of the liquid to infiltrate the surface or its interaction with the solid surface. Wettability can vary in different environments. Surface treatments, chemical modifications, or external factors can alter it. This variability is significant as it impacts various aspects, such as whether a liquid infiltrates and spreads across a surface or, conversely, maintains a distinct shape. It also plays a role in enhancing heat exchange capabilities.

Environments in which liquids flow or descend onto solid surfaces are common in everyday life [20,21]. In this study, we focused on the wettability of the dispenser nozzle. Many manufacturing processes in modern industry involve the dispensing of various liquids, which often require dispensers. Despite the development of dispensers, only a few types of needles are capable of passing liquids through the final stage of dispensers [22]. STS304 is commonly used as the material for dispensing needles in the dispenser industry. It is a cost-effective material that exhibits minimal reactivity with substances, making it suitable for dispensing needles [23]. Recently, there has been an increasing demand for high precision in processes, particularly in industries such as semiconductor and pharmaceutical companies, where extremely precise procedures are essential [24]. Unfortunately, STS304 falls short of meeting the requirements for such precision processes. To overcome this limitation, various methods have been proposed, and one notable solution is coating the surface of the needle. Coating involves layering the surface of the needle with different materials to impart specific properties. Through this process, attributes like surface hardness, friction coefficients, and wear resistance can be adjusted. Coating the surface of the needle can enhance its performance in precision processes. Additionally, careful selection of coating materials and techniques can optimize the performance for specific applications within the needle’s intended use. This approach offers a means to overcome the shortcomings of STS304, providing a solution to meet the increasing demand for high precision in various industries.

The coating material possesses hydrophobic properties that can render the needle surface in a lower energy state than its original state [25]. This means that when a liquid comes into contact with the needle surface, it does not linger but easily flows off, facilitating precise flow during dispensing processes. Furthermore, a hydrophobic surface on the needle, particularly in dispensing highly viscous liquids, prevents the liquid from adhering to the external surface of the needle during ejection, minimizing the formation of tails or satellite droplets [26,27]. Coatings with such properties are widely used across various industries to optimize dispensing needle performance. Therefore, they can be effectively employed in applications where precise control of liquids is required. However, coating involves the application of different substances onto the STS304 surface, and over time, it can be affected by external environmental factors. Specifically, depending on external conditions and usage frequency, the hydrophobic properties of the coated surface may diminish after a certain period [28]. As a result, periodic recoating may be necessary for coated needles, depending on the usage environment and exposure conditions. Moreover, if the coating wears off, restoring the surface to its original properties can be challenging, making the management of the service life a crucial consideration. These limitations need careful consideration, especially in industries where precise control and durability are essential, highlighting the need for periodic maintenance and attention to the service life of coated needles in specific applications. Nevertheless, despite its drawbacks, the coating process has clear advantages compared to untreated surfaces. Many researchers are actively studying the types of coating materials and coating methods to mitigate these disadvantages.

Zhichao and Dong’s study integrated surface structures with coatings [29]. Coating was applied to surfaces after the formation of structural elements, altering their wettability. This demonstrated the potential to enhance dispensing accuracy and reduce costs. However, since this involves two separate processes—forming surface structures and coating—effectively offsetting the drawbacks of coating is expected if precise dispensing can be achieved without using additional materials in their original state. In this study, attention was focused on this aspect, aiming to enhance the performance of STS304 needles by forming fine porous structures on the needle surface without coating. An important consideration is to uniformly create an environment where structural elements are formed on the needle surface to maintain a consistent surface state. In this research, a solution similar to Zhichao and Dong’s study was used, but the ratio was adjusted, and electrochemical etching was employed to form structures by passing current. No additional coating was applied after structure formation. To analyze the performance of the created needles, various liquids with different properties were dispensed. The resulting phenomena were observed and analyzed to explore the feasibility of substituting coated needles for precision dispensing.

## 2. Experimental Methods

### 2.1. Fabrication of Microstructures via Electrochemical Etching

As reported by Cassie and Baxter [30], microstructures on metal surfaces cause wettability changes. In this study, microstructures were created via electrochemical etching to change the wettability of the STS304 needle surface. Before etching, the surface was polished to remove impurities and washed with an ultrasonic cleaner. The liquid remaining after the ultrasonication was evaporated using a furnace. Aqua regia, a mixture of 2% HCl and 1.2% HNO_3_, was considered suitable for etching a surface structure on the 21G needle used in the experiment. The electric field generated when a cathode and an anode were formed in the solution formed a structure on the surface; a voltage was further applied [31]. A power supply was applied to the positive and negative electrodes for etching. The STS304 needle, in which the structure was formed, became the cathode, whereas the carbon sheet, which facilitated the formation of an electric field, was connected to the anode.

The needle and carbon sheet were placed perpendicular to each other to form a 360° structure in the same environment on the needle surface. A copper rod and a metal clamp were used to facilitate the flow of electricity through both the needle (anode) and the carbon sheet (cathode). In this case, the white part of the metal rod was coated to ensure that it did not interfere with the etching process (Figure 1a), and only the part in contact with the carbon sheet remained uncoated. Current was not allowed to flow through the masked section (Figure 1b) except on the side that was in contact with the carbon sheet. Typically, when a large amount of heat is generated during etching, the reaction becomes stronger and destroys the needle surface [32]. Therefore, a constant-temperature water tank was used to control heat, and the temperature was set to 10 °C. Etching was performed for 3 min in the experimental setting depicted in Figure 1c; a voltage of 3.2 V suitable for forming a surface structure was applied to the power supply.

### 2.2. Dispensing Liquid Droplet Test Setup

To determine the effect of the surface structure of the etched needle, we used an etched needle and a non-etched needle to dispense the liquid at 0.9 μL/min. Surface tension was the variable assumed to influence liquid dispensing owing to the surface structures. Accordingly, solutions were prepared using distilled water and 99.9%, 90%, 80%, and 10% methanol [33,34,35,36,37,38]. The methanol solutions used in the experiments, with concentrations of 90%, 80%, and 10%, were prepared based on volume fractions. The ejected liquid droplet at 0.9 μL/min was photographed using a high-speed camera (FASTCAM Mini AX, Photron, Tokyo, Japan.) with a frame number of 10,000 frames/s. A white cloth was placed behind the needle to serve as the background, and the lighting was directed onto it to capture photographs. The images were captured at a location 2 cm away from the needle tip (Figure 2). All experimental devices were configured such that the surroundings were blocked with plastic walls and ceilings (Figure 3) to minimize the influence of the external environment. The laboratory temperature and humidity were maintained at 25 °C and 52–53%, respectively. The images were analyzed using a digital image processing script in MATLAB^®^ (R2021b) to determine the ejected droplet velocity and diameter.

## 3. Results

### 3.1. Results of Etching

In this study, porous structures were fabricated via electrochemical etching. Figure 4a depicts the non-etched area, which comprised a few scratches on the surface. A porous structure was formed in the etched area, as indicated in Figure 4b. As explained in Section 2.1, several variables, such as the concentration, applied voltage, and reaction temperature, contribute to the formation of these structures. However, if the reaction is excessively strong, as illustrated in Figure 5, the surface may be damaged, as indicated by the red dashed lines in both Figure 5a,b, or surface structures may not form. In Figure 5a, the red dashed lines represent the appearance of the needle tip breaking due to the strong reaction, and in Figure 5b, the surface peeling is observed after the formation of porous structures. As evident in Figure 5b, such peeling results in a reduction in the thickness of the needle. On the other hand, if the reaction is considerably weak, the formed structures may be small and not uniformly distributed. In order to create appropriate structures, a series of procedures may be necessary.

Figure 6 illustrates the experimental results obtained from the etching method described in Section 2.1. Figure 6a indicates that porous structures are formed on the exterior, and structures are also formed at the end of the needle, similar to the green region in Figure 6b. The red region in Figure 6c represents the appearance of the internal structures, which are not porous yet exhibit a consistent pattern.

To compare the differences between the etched and non-etched needles, a low-viscosity and low-surface-tension substance, i.e., methanol, was dispensed (Figure 7). Initially, the dispensed methanol spread across the surface of the needle, with certain differences in terms of the direction of spreading between the etched and non-etched needles (Figure 7a,b). This difference in the direction of spreading affected the direction in which the droplets descended. A conventional needle tends to spread the droplet significantly in specific directions, resulting in uneven spreading of the liquid on the external surface of the needle. By contrast, the etched needle exhibited a relatively uniform spreading pattern without significant bias in specific directions. This was because the evenly distributed porous structures formed via etching altered the surface wettability of the needle. Figure 8 shows the differences in the droplet dispensing process at the needles.

Furthermore, certain changes were introduced to the etched area. The first case involved etching only the extent of the area where the liquid spreading was observed on the conventional needle, whereas a wider area than that of the typical spreading area of the liquid was etched in the second case. No significant differences were observed in the results between the two cases. For instance, the degree of liquid spreading was identical, and even when a sufficient area was available for the liquid to spread, the spreading was biased toward a specific direction if the etching was not uniform in all directions. Therefore, we concluded that the porous structures should be formed in an area that matches the degree of liquid spreading observed on a conventional needle with uniform etching in all directions of the needle.

### 3.2. Results of Dispensing

As explained in Section 3.1, the results show that methanol droplets formed in a uniform direction on the etched needle. We presumed that the main parameter affected by dispensing was surface tension. Therefore, two liquids with different surface tensions (water and 99.9% methanol) were ejected for comparison; Figure 9 illustrates the corresponding results. The actual volume values of all dispensed liquids, as seen in Table 1, vary for each liquid. Therefore, the vertical axis represents the percentage obtained by averaging the volume values per droplet and expressing them as a percentage, while the horizontal axis represents each individual droplet.

In the case of methanol, the standard deviations in the etched and non-etched needles were 0.00539 and 0.01747, respectively. It was evident that the droplets ejected from an etched needle exhibited more stable values that were close to the mean value compared to those ejected from a non-etched needle, wherein the deviation from the mean value was significantly high. In other words, etching resulted in a more stable ejection. Even when dispensing water, the standard deviations for the etched and non-etched needles were 0.00248 and 0.00257, respectively. These results confirmed that the ejection of water was more stable when using etched needles in comparison with non-etched needles; however, the difference was not significant compared with 99.9% methanol.

To examine the ejection tendencies in liquids with various surface tension values other than water and methanol, we ejected 90%, 80%, and 10% methanol in the same manner. As mentioned in Section 3.1, methanol exhibits a spreading phenomenon over the needle surface. This phenomenon occurs due to the low surface tension. However, for methanol 80% and 10%, which have slightly higher surface tension, the spreading phenomenon did not occur when ejecting the liquid, unlike in the case of 99.9% methanol. Both liquids, as evident in Figure 10, showed that the ejection from etched needles in 80% methanol had a standard deviation of 0.01197, and in 10% methanol, it was 0.00819. Comparing these values with the standard deviations of 0.01006 and 0.00533 from non-etched needles when ejecting liquids, both showed significantly larger standard deviations. In the case of 90% methanol, the standard deviation of the mean volume from etched needles was 0.00776, while from non-etched needles, it was 0.04503. In this case, the etched needles resulted in a smaller value. Considering the significantly higher surface tension of water compared to methanol, the larger standard deviations observed when 80% methanol and 10% methanol were etched may be attributed to errors such as air resistance or the pump pushing the liquids when the droplet falls. Consequently, it can be inferred that, especially when the surface tension is low, a more stable ejection volume is achieved.

We also compared the average ejection speeds, and as evident in Figure 11, for water with the highest surface tension, the standard deviation from etched needles was 0.00235, while from non-etched needles, it was 0.00392, indicating a lower value when etched. For 99.9% methanol, which has the lowest surface tension, the standard deviation from etched needles was 0.00206, while when not etched, it was 0.01351, showing a difference of over six times. Both 90% and 80% methanol, when etched, had lower standard deviations, with values of 0.00542 and 0.00507, respectively, compared to the non-etched standard deviations of 0.00819 and 0.00551. For 10% methanol, only when etched, a standard deviation of 0.00500 was observed, while when not etched, it was 0.00415, resulting in the opposite outcome. Similar to the volume comparison, considering the significantly higher surface tension of water, these variations seem to be due to experimental errors. It can be noted that 99.9% methanol, with the lowest surface tension, exhibits a very stable ejection speed.

## 4. Discussion

In this study, microstructures were formed on needle surfaces via electrochemical etching. To perform the etching, it was essential to establish variables such as the appropriate solution ratio, reaction temperature, and applied voltage. Additionally, a uniform arrangement of microstructures in all directions of the needle was achieved by ensuring a perpendicular alignment of the cathodic needles and anodic carbon plates. Due to the risk of needle breakage, surface oxidation, and delamination with excessive reaction intensity, a set of recipes with specific variables for each needle size was crucial. In the experiments, an appropriate porous structure was successfully created using 21G as the reference size. Subsequently, the needles with formed structures on their surfaces were then employed to dispense various types of liquids, allowing the observation of how the presence or absence of structures influenced the liquid behavior. Liquids with low density and surface tension tended to spread well on surfaces when dispensed. Moreover, they spread on the surface before falling, rather than immediately ejecting from the needle tips. The direction and extent of spread were more consistent when the needles were etched. This phenomenon was observed for both 99.9% and 90% methanol. By contrast, water, 80% methanol, and 10% methanol did not spread on the needle surfaces. The visual changes in appearance could not be distinguished by the naked eye when using etched needles, particularly when the dispensed liquids did not spread on the surface. Therefore, high-speed cameras were used to capture the process, and the obtained images were analyzed.

First, we compared the average volumes of droplets using 99.9% methanol and water, taking into account their different surface tensions. In both cases, when etched, a more stable ejection was observed, as indicated by the lower standard deviation in the average volume of the falling droplets. To assess liquids with a broader range of surface tensions, we ejected 90%, 80%, and 10% methanol. The analysis showed that for 80% and 10% methanol, etched needles resulted in a higher standard deviation. However, considering that both liquids have lower surface tensions than water, these results are within the expected error range. We also compared the ejection speeds in a manner similar to the volume comparison. Except for 10% methanol, when all were etched, the standard deviations were lower. This result, like the volume comparison, falls within the expected error range, and it indicates that lower surface tension leads to a more stable ejection speed.

The consistent volume of the falling droplets implies a constant amount of dispensed liquid, and a steady ejection speed is valuable in processes where a uniform dispensing rate is required. When dealing with liquids prone to external spreading due to low surface tension, using etched needles as the operating liquid can result in consistent dispensing despite the frequent occurrence of spreading phenomena. This suggests the potential to replace coated needles with a shorter lifespan. Moreover, by substituting coated needles, economic losses arising from the short lifespan of coated needles can be mitigated.

In this study, we observed that even liquids with low surface tension tend to be dispensed reliably. However, when ejecting liquids with high viscosity, such as glycerol, which has a lower surface tension than water, we could not confirm stable ejection. This remains a subject for future research, and subsequent studies should aim to demonstrate the effectiveness of etched needles across various liquid properties.

## 5. Conclusions

This paper is written to address the difficulties in ensuring precision in dispensing needles commonly used in industry, considering the short lifespan of coated needles that have been developed to overcome the issues. In order to secure a longer lifespan without coating the needles with other substances, a micro/nanoporous structure was created on the needles. This addresses the problem of imprecision caused by the phenomenon where liquid wets the external surface of the needle when dispensing low-surface-tension liquids, a problem inherent in uncoated needles. The surface structure can be formed using the electrochemical etching method with concentrated sulfuric acid. Still, there were many variables to consider, such as reaction temperature, solution concentration, and reaction time. When using STS304 21G needles under the condition of 10 °C with an applied voltage of 3.2 V and a reaction time of 3 min, an appropriate micro-/nanoporous structure was formed. The needles with the formed micro/nanoporous structure (etched needles) exhibited a consistent level of spreading on the outer surface when dispensing liquids with low surface tension. Moreover, they showed better directional ejection compared to the non-etched needles. As a result of comparing the average volume and average speed by dispensing several liquids with different surface tensions, both 99.9% methanol with the lowest surface tension and water with the highest surface tension were dispensed at a lower standard deviation from the etched needle, which means that the etched needle has a stable dispense. In addition, the lower the surface tension, the greater the degree to which the etched needle has a stable dispense than the needle that does not. It was observed that as the surface tension decreases in etched needles, a consistent amount and speed of dispense can be achieved. However, the drawback of the occurrence of tail and satellite droplets when using high-viscosity liquids as the working liquid could not be resolved. This is an issue that can be addressed when using coated needles, and for practical industrial applications of etched needles, further research is needed to make them compatible with a wider range of liquids.

## Figures and Tables

**Figure 1 micromachines-14-02183-f001:**
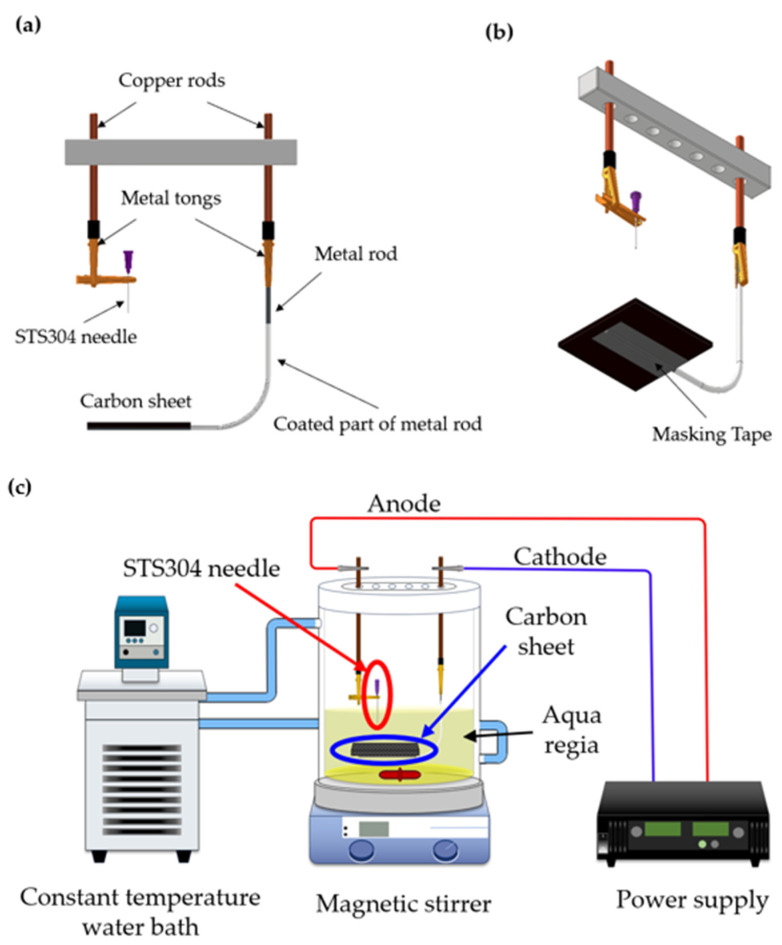
Structure for etching (**a**) front view, (**b**) bottom view, and (**c**) entire electrochemical etching setup.

**Figure 2 micromachines-14-02183-f002:**
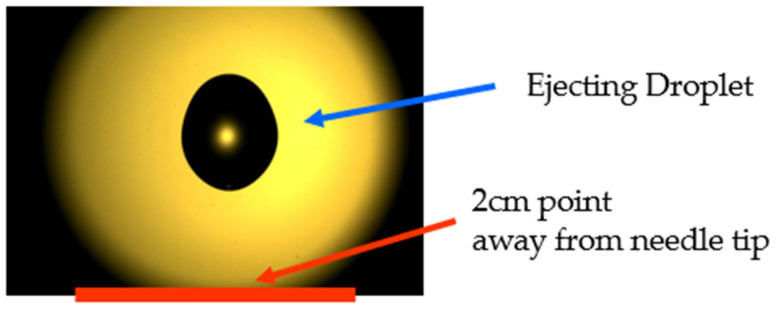
Captured image of the ejected liquid droplet.

**Figure 3 micromachines-14-02183-f003:**
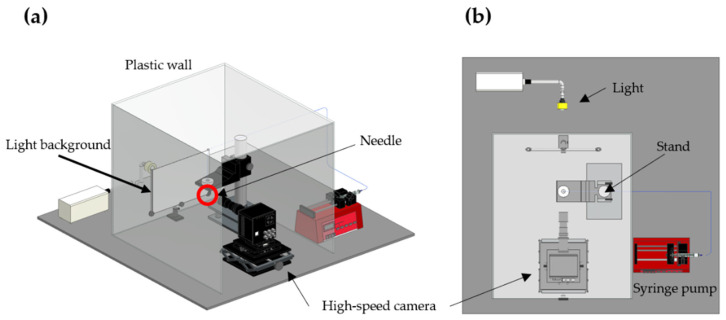
Dispensing test setup’s (**a**) perspective from the diagonal and (**b**) overhead view.

**Figure 4 micromachines-14-02183-f004:**
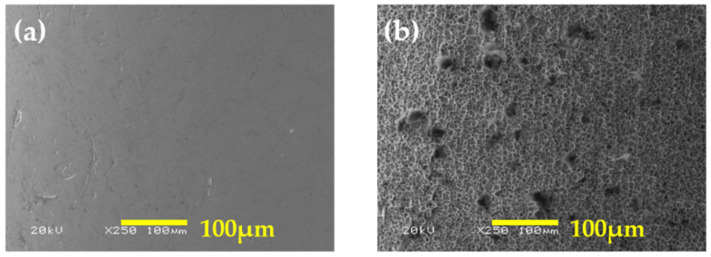
Surfaces of the (**a**) non-etched and (**b**) etched needles.

**Figure 5 micromachines-14-02183-f005:**
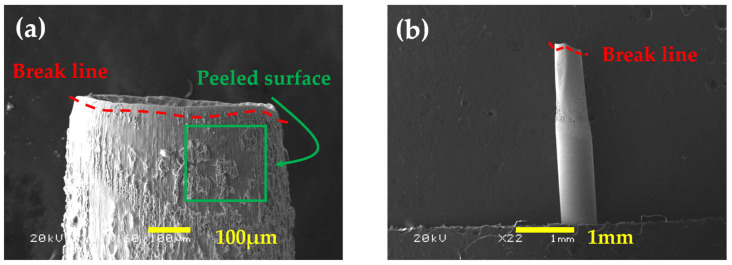
Needles (**a**) peeled and (**b**) broken by a strong reaction.

**Figure 6 micromachines-14-02183-f006:**
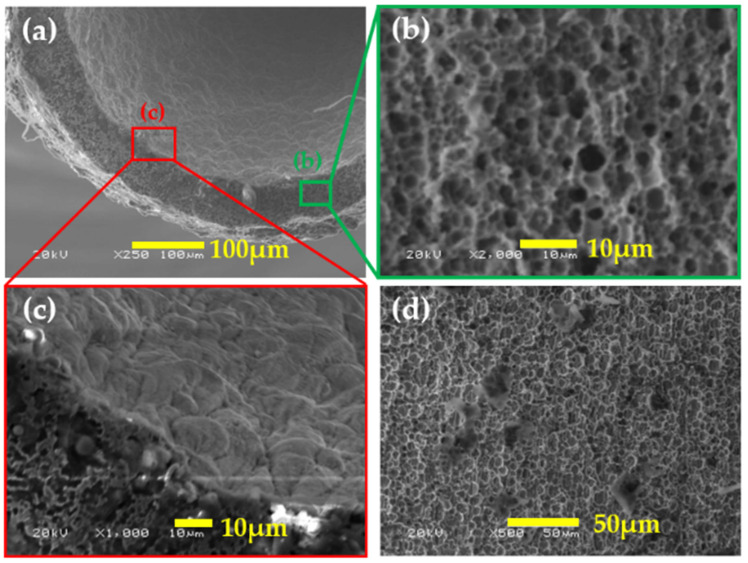
The (**a**) inside, (**b**) tip, (**c**) structure of the inside, and (**d**) external surface of the etched needle.

**Figure 7 micromachines-14-02183-f007:**
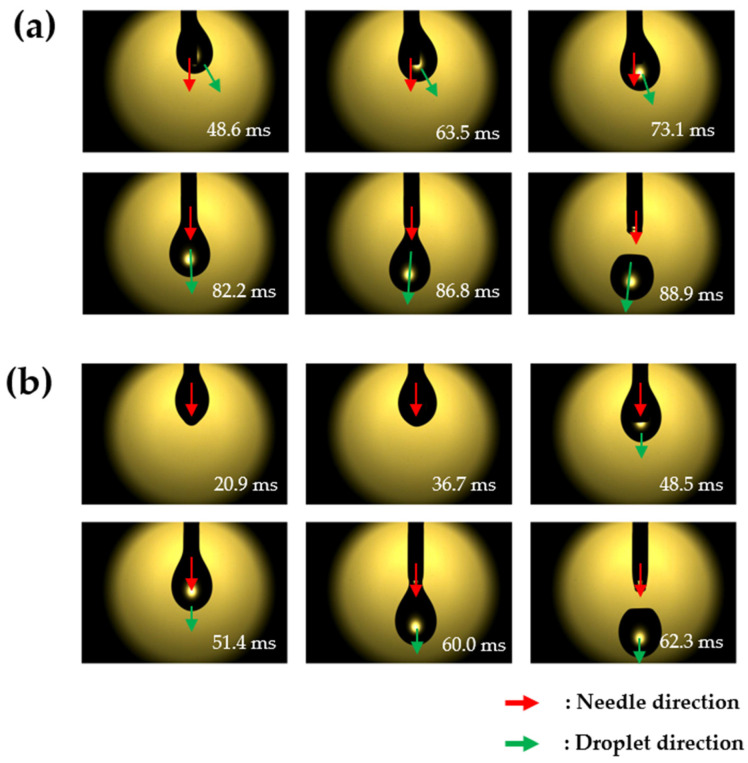
Droplets dispensed from the (**a**) non-etched and (**b**) etched needles.

**Figure 8 micromachines-14-02183-f008:**
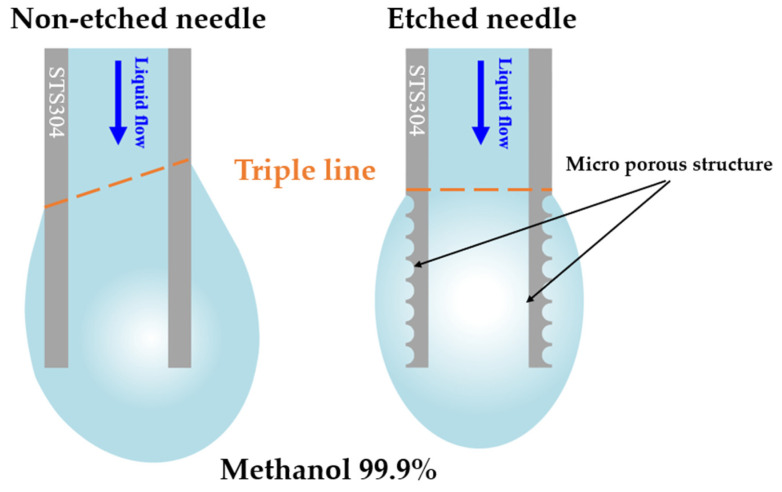
A schematic of different droplet formations at the needles.

**Figure 9 micromachines-14-02183-f009:**
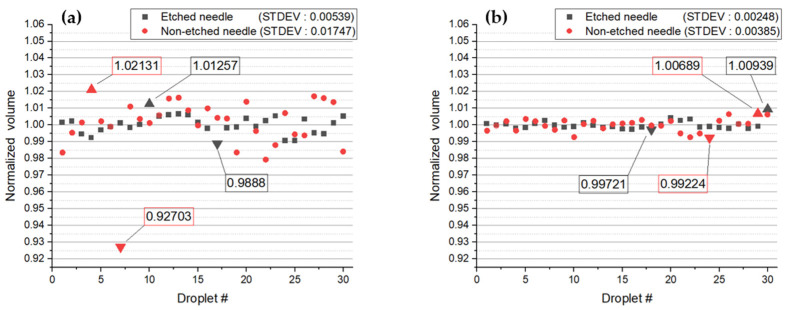
The volume of each droplet: (**a**) 99.9% methanol; (**b**) distilled water.

**Figure 10 micromachines-14-02183-f010:**
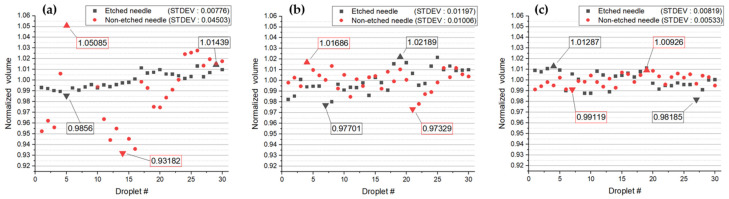
Volume of each droplet: (**a**) 90% methanol; (**b**) 80% methanol; (**c**) 10% methanol.

**Figure 11 micromachines-14-02183-f011:**
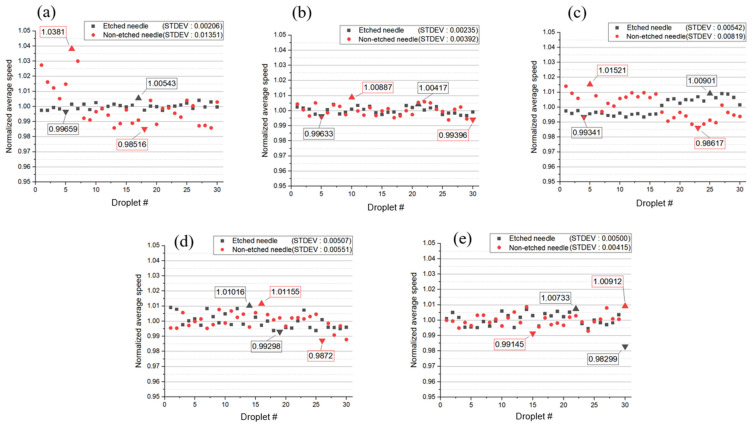
Velocity of each droplet: (**a**) 99.9% methanol; (**b**) distilled water; (**c**) 90% methanol; (**d**) 80% methanol; (**e**) 10% methanol.

**Table 1 micromachines-14-02183-t001:** The range of actual volumes ejected from each needle.

Liquid	Etched Needle	Non-Etched Needle
99.9% Methanol	6.09~6.24 µL	5.46~6.01 µL
90% Methanol	6.42~6.61 µL	5.88~6.88 µL
80% Methanol	6.59~6.89 µL	6.11~6.38 µL
10% Methanol	7.84~8.09 µL	10.41~10.60 µL
Water	11.11~11.25 µL	14.95~15.17 µL

## Data Availability

The data supporting the findings of this study are available from the corresponding author upon request.

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
