# Peer review of "Performance Improvement of an STS304-Based Dispensing Needle via Electrochemical Etching"

_micromachines, 2023, doi:10.3390/mi14122183_

Round 1

Reviewer 1 Report

Comments and Suggestions for Authors

The authors electrochemically etched to form microporous structures on the surface of the STS304-based dispensing needle for huge improvement performance compared to conventional needles. They observed that the wettability of the surface changed because of the microporous structures when different liquids with varying properties were discharged from the needle, as well as examined the possibility of replacing the coating of the needle with porous structures to enable precise discharge. This confirms that liquids within a certain viscosity range benefit from the porous structures created on the needle's surface. The experimental results showed that etched needles require the ability to effectively discharge a wider range of fluids. Meanwhile, the etching can extend the lifespan of surfaces as it does not involve the coating of additional substances. These characterizations will find a place in a new category of dispensing needles. This manuscript has certain value and can be used as a reference for subsequent research workers to carry out relevant research but still requires some minor revisions before it can be considered for publication.

1.    To facilitate reader comprehension, it is suggested that the authors improve the quality of the manuscript's writing.

2.    In the Introduction section, there are few descriptions of relevant research background. Furthermore, more discussions on the relevance of these references the present research needs to be provided.

3.    In the Results section, when comparing the differences between etched needles and non-etched needles, it is recommended that the author increase the readability of the article by using Tables and discuss the surface micromorphology, coating appearance, roughness, adhesion, etc.

4.       It is recommended to briefly summarize the manuscript's research and provide a full discussion of the study's findings. Additionally, we encourage the authors to add more information on the application prospects of etching STS304-based dispensing needles.

Author Response

Thank you for your kind reviews and comments to the paper “Performance Improvement of an STS304-based Dispensing Needle via Electrochemical Etching”. We carefully considered each of the comments and revised the paper according to them. In the revised manuscript, the changes reflecting the revision are marked in red.

Reviewer 2 Report

Comments and Suggestions for Authors

The authors present an original, recent and interesting document.

However, the range and nature of the liquids used is small (despite the combinations/mixtures presented, e.g. 90%, 10%). It is important that the authors improve the document by including at least one liquid (and corresponding combinations) with a polarity more distant from that of methanol and water. 

I note that the authors have tested glycerol. However, due to the high viscosity of the chosen liquid, the test did not allow the necessary conclusions to be drawn for a work of this nature. This approach will only be viable if glycerol is diluted (different percentages). 

Alternatively, the authors should choose another less polar liquid and repeat the tests.

In section 3.1, it is important that the authors provide a picture to illustrate the appearance of the surface when it has been damaged by the intense reaction.

Author Response

(The authors gave the same response as above.)

Round 2

Reviewer 1 Report

Comments and Suggestions for Authors

accept

Reviewer 2 Report

Comments and Suggestions for Authors

The authors have made changes to the manuscript to incorporate the reviewer's comments. They have also explained why, in some cases, the reviewer's suggestions are not reflected in the new version of the manuscript.

I therefore believe that the manuscript is acceptable for publication.